# Repeatable Crack Self-Healing by Photochemical [2 + 2] Cycloaddition of TCE-co-DCE Monomers Enclosed in Homopolymer Microcapsules

**DOI:** 10.3390/polym11010104

**Published:** 2019-01-09

**Authors:** Sunyoung Kim, Bo-Hyun Kim, Myongkeon Oh, Dong Hyuk Park, Sunjong Lee

**Affiliations:** 1Korea Institute of Industrial Technology, Chenan, Chungnam 31056, Korea; sykim0722@kitech.re.kr (S.K.); bohkim@kitech.re.kr (B.-H.K.); arudy321@kitech.re.kr (M.O.); 2Department of Applied Organic Materials Engineering, Inha University, Incheon 402-751, Korea

**Keywords:** crack-healing, self-healing, photochemical 2 + 2 cycloaddition, homopolymer microcapsules

## Abstract

Self-healing, an autonomous repairing process stimulated by damage, has recently attracted a great deal of attention in the field of medical and mechanical engineering as well as from scientists, due to its valuable potential applications. However, as the self-healing process is mediated by specific functional materials, practical applications have been limited. Here, we introduce a healable homopolymer microcapsule that can self-heal a crack or cleaved part through a photochemical [2 + 2] cycloaddition process. Microcapsules were prepared through photopolymerization and suspension polymerization, each containing 1,1,1-tris (cinnamoyloxymethyl) ethane (TCE) and 1,1-di (cinnamoyloxymethyl) ethane (DCE) monomers, which act as healing materials. TCE and DCE monomers were polymerized into poly (TCE-co-DCE) without a photoinitiator under illumination. The epoxy specimen embedded with microcapsules showed obvious healing performance during illumination after cracking. From the FT-IR spectra for each step of the healing process, the specimen could be repeatedly self-healed through the reversible process of cyclobutane cross-links to the original cinnamate and vice versa. This work shows an alternative approach using homopolymer microcapsules to accomplish the repeatable self-healing of a crack without interface discontinuity, which could be adopted as a healing substance in various paints.

## 1. Introduction

In the past decade, major advancements in ‘‘smart’’ polymeric materials have been made in the field of shape memory polymers [1] or adaptive and responsive polymer surfaces [2]. The common theme among these materials is that they respond autonomically to the damages through thermal, mechanical, ballistic or other ways [3]. The fatigue in plastics caused by a steady or cyclic stress results in the generation and propagation of microscopic crack, eventually leading to a significant failure in the mechanical performance of the materials. Therefore, much attention has been paid to the treatment of cracks in polymer matrices. Self-healable polymers are another type of recently developed smart material. This smart polymer is able to actively cure minor damage, acting as a stimulus to initiate the process of autonomically repairing the damage [4].

So far, some self-healing techniques have been reported: encapsulation of an instant healing agent such as cyanoacrylate in the microcapsule, a vascular network approach using a tube containing or flowing a healing agent, and intrinsically healable materials that work with external stimuli. Among them, the microcapsule filled with various liquid healing agents is a successful and versatile approach, although the performance and repeatability of healing still need further enhancement [5,6,7,8]. While propagating a crack induced by external stress, microcapsules are ruptured and release a healing agent. The released agent fills the damaged area and then starts the curing process when exposed to light or heat. As self-healing with microcapsules cannot restore the damaged area to its original state, it is used for fiber-reinforced composites composed of different thermosetting polymeric materials. However, this encapsulation method is only applicable to one-time healing, and is not repeatable due to the different materials between the core therapeutic agents and the shell of the microcapsules. In addition, because the fine cracks existing between the isolated microcapsules can not recover, the synthesis of smaller and smaller microcapsules (<10 μm) has been developed. However, when using smaller microcapsules, the weight fraction of healing materials needs to be increased for efficient self-healing [9,10,11,12]. Accordingly, to expand the limit of the microencapsulation technique, new types of microcapsule and core healing agents should be introduced for repeatability without leaving gaps.

In this study, we suggest a system of microcapsules consisting of 1,1,1-tris (cinnamoyloxymethyl) ethane (TCE) and 1,1-di (cinnamoyloxymethyl) ethane (DCE) monomers as core healing agents and a homopolymer shell made of (TCE) and (DCE). This particular chemical system was developed to overcome the limitation posed by one-time healing. We tried to form core-shell-like microcapsules by exposing them to daylight or UV (≥280 nm) following suspension polymerization, while the core portion remained as a monomer due to the limit of depth of light penetration. From the devised damage of epoxy specimens, which were embedded with microcapsules, we examined their healing performance and repeatability using scanning electron microscope (SEM), FT-IR and fluorescence microscope images. 

## 2. Materials and Methods

### 2.1. Materials

1,1,1-(hydroxymethyl) ethane (99% purity, Aldrich, St. Louis, MO, USA), 2-methyl-1,3-propandiol (99% purity, Aldrich, St. Louis, MO, USA), cinnamoyl chloride (Aldrich, St. Louis, MO, USA), 4-(dimethylamino) pyridine (DMAP, 99% purity, Aldrich, St. Louis, MO, USA), sodium p-styrene sulfonate (NaSS, Aldrich, St. Louis, MO, USA) and sodium p-styrene sulfonate were used as received without further purification, and were kept at −5 °C until use. Tetrahydrofuran refined class reagent (THF, J.T. Baker, Pennsylvania, USA) was used for the synthesis of the monomer. Distilled and deionized (DDI) water was used throughout the experiment. EMbed-812 (Epon-812 Substitute) resin, nadic methyl anhydride (NMA) (methyl-5-norbornone-2,3-dicarboxilic anhydride) cross-linker and Tri (dimethylaminoethyl)phenol (DMP-30) (Electro Microscopy Sciences. CO., LTD., Hatfield, PA, USA) for an acceleration agent were used for the matrix.

### 2.2. Synthesis of DEC-co-TCE Monomer

1,1,1-(hydroxymethyl) ethane 1.74 g (1.4 × 10^−2^ mole), 2-methyl-1,3-propandiol 0.57 g (6.3 × 10^−3^ mole) and 4-(dimethylamino) pyridine 6.60 g (1.8 × 10^−2^ mole) were added to the tetrahydrofuran 60 mL and completely dissolved under a N_2_ atmospheric condition. After that, the cinnamoyl chloride 9.00 g (5.4 × 10^−2^ mole) dissolved in 15 mL of the tetrahydrofuran was slowly dropped into the mixed solution and refluxed at 80 °C for 5 h. During this reflux process, tetrahydrofuran was completely removed by distillation and the methylene chloride, which is the byproduct of the reaction, was extracted. After completion of the reaction, the final product was purified using a 0.5 M aqueous NaOH solution three times and then separated into TCE and DCE by a column using 1:1 n-hexane and ethyl acetate.

### 2.3. Preparation of Poly (TCE-co-DCE) Microcapsules and Epoxy Specimen

To fabricate poly (TCE-co-DCE) microcapsules, we consecutively adopted particle colloid synthesis and photopolymerization methods. The ratio between TCE and DCE monomers was varied as 90:10, 70:30, 50:50, 30:70, and 10:90. For the suspension polymerization, a 20 wt% of (TCE-co-DCE) monomer and a 20 wt% monomer of NaSS were added to the distilled and ionized (DDI) water, and then homogenized at about 10,000 rpm for 5 min (IKA ULTRA-TURRAX homogenizer). Right after the suspension polymerization, UV light (main λ = 365 nm) was irradiated for 30 min (Spot UV Cure System INNO-CURE 850–1200 W) while the solution was stirred at 800 rpm using a magnetic stirrer. After microcapsule polymerization, it was washed three times with DDI water and dried in an oven (80 °C) for 24 h. Test specimens were prepared using two kinds of epoxy-based resins, EMbed-812 (Epon-812 substitute) and DMP-30 {tri(dimethyllaminoethyl)phenol}, where nadic methyl anhydride (NMA) (methyl-5-norbornone-2,3-dicarboxilic anhydride) was used as a cross-linker and acceleration agent. The dried poly (TCE-co-DCE) microcapsules were sufficiently mixed with epoxy-based resin and cross-linker. The mixed resin was inserted into Teflon molds and irradiated for 10 min with a wavelength (λ) > 280 nm.

### 2.4. Characterization

For the analysis of morphology, FE-SEM (JSM-600F, JEOL, Seoul, Korea) was used. The FT-IR spectra were obtained using a Tensor 27 (BRUKER, Seongnam-si, Korea). Fluorescence images were recorded with a spectrofluorophotometer (RF-5301PC, Shimadzu, Seoul, Korea). Proton nuclear magnetic resonance (^1^H NMR) was used for the structural analysis of the monomer.

## 3. Results

### 3.1. Synthesis of DCE, TCE and Poly (TCE-co-DCE)

Figure 1a schematically shows the molecular structure of TCE and DCE synthesized by the process (see details in Section 2). According to the process, the yield of TCE and DCE co-monomer in the final process was 70%. From the ^1^H NMR spectrum of the TCE and DCE co-monomer, the characteristic peaks of TCE and DCE structures were confirmed with extremely low sub-peak intensities from impurities. We confirmed the chemical structure of TCE and DCE co-monomer using the ^1^H NMR spectrum shown in Figure 1b. Using the TCE-co-DCE monomers, the poly (TCE-co-DCE) was synthesized. Each signal in the ^1^H NMR spectrum of the TEC-co-DCE monomers can be assigned to each proton on the chemical structure. The signal at 4.29 ppm numbered as 6 is the trace of OCH_2_, supporting that the cinnamoyl was successfully binding with tris(hydroxymethyl) ethane. After the synthesis of TCE-co-DCE monomers, we tried to polymerize them using UV irradiation. Figure 1c shows the comparison of FT-IR spectra after (1) and before (2) the completion of polymerization as a thin film. In the spectrum of 2, there are two characteristic absorption bands. One is the band peak at 1713 cm*^−^*^1^ and the other is at 1637 cm*^−^*^1^, which can be assigned to C=O and C=C vibrations of the cinnamoyl group, respectively. The common peaks at 1578 cm*^−^*^1^ and 1604 cm*^−^*^1^ are assigned to aromatic C=C stretching and cyclobutane double bond stretching. Meanwhile, spectrum 1 shows only one characteristic band peak at 1734 cm*^−^*^1^, which can be assigned to the peak shift of carobnyl absorption. The band for C=C absorption is shown to be almost disappeared in spectrum 1. This change of FT-IR means that while irradiated by light, TCE-co-DCE monomers were photo-cross-linked to be poly (TCE-co-DCE). Based on the peak intensity, the intermolecular cross-linking of cinnamoyl groups occurred up to ~99% after 120 s of irradiation. However, during this process intramolecular cyclization could also be undertaken due to the possibility of the intramolecular cycloaddition of propylene 1,3-dicinnamate. Figure 1d shows the optical and fluorescence images of TCE-co-DCE thin film observed during UV irradiation. The left is an optical image showing fine wrinkles and valley-like tearing lines. The right is a fluorescence image of the same area shown on the left. The fluorescence on the valley-like tearing lines can be clearly observed, in contrast to the other area. This fluorescence originates from the TCE-co-DCE monomers, showing whether the monomer is spilled out of the microcapsules and polymerized under the illumination of light.

### 3.2. Core-Shell-Like Microcapsules

Core-shell-like microcapsules were synthesized by using consecutive suspension polymerization and photopolymerization methods, which is a new trial, not yet reported. In this process, homopolymer microparticles are firstly fabricated by suspension polymerization in the water phase and simultaneously exposed to daylight or UV (≥280 nm) for the photochemical 2 + 2 cycloaddition, as schematically shown in Figure 2a. The advantages of the developed process are the abilities to make a spherical-shaped particle by suspension polymerization and a core-shell-like microparticle containing the monomers similar to the shell by photopolymerization. Also, the shell thickness can be controlled by the UV wavelength because of the penetration depth of light. However, due to the high viscosity of the mixture in the water phase, which disturbs the homogeneous dispersion of monomers, stabilizer and agent, there was a large variation of particle size. According to the ratio of TCE to DCE, varying from 90 (#1) to 70 (#2), 50 (#3), 30 (#4) and 10 (#5), the solution viscosity and the size of microcapsules were changed (Figure 2b,c). After the UV irradiation, the polymerized colloidal aqueous solutions were cloudy like milk. The inset SEM images of Figure 2b show synthesized poly (TCE-co-DCE) particles with the different TCE ratios. As the ratio of TCE decreased from 90 to 10, the size of the microcapsules decreased from ~10 µm to ~0.5 µm, which is in line with the monomer viscosity listed in Table 1. It is well known that when using smaller microcapsules, the weight fraction of healing materials should be increased for efficient self-healing. Based on this result, the appropriate ratio of TCE to DCE was 70 to 30 (#4), where the viscosity was 4457 cP, and the particle size was relatively homogeneous with about 1 µm.

### 3.3. Cracking and Crack-Healing Performance of Microcapsules

The crack-healing ability of TCE-co-DCE microcapsules was examined. Figure 3a shows the poly(TCE-co-DCE) microcapsules before being embedded in the epoxy specimen and Figure 3b shows the broken microcapsule and the monomers spilled out of the microcapsule. After being broken, the spherical shape of the microcapsule was shrunken due to the outflow of monomers. The area covered by spreading monomers indicated by the white-dashed line became smooth after being illuminated with UV light. Figure 3c shows a microcapsule-embedded epoxy specimen. Test specimens with microcapsules were prepared using two kinds of epoxy-based resins through the thermal polymerization method. Figure 3d shows the microtoming cross-section of the epoxy specimen with an embedded microcapsule. The area of the microcapsule looks darker than the epoxy and it is also smooth compared to the area of Figure 3b after UV exposure. This means that the TCE-co-DCE monomers enclosed by poly(TCE-co-DCE) capsules were well polymerized even after being embedded in the epoxy.

This prepared specimen was inserted between two stainless steel plates, and cracks were created by tapping the banding one or two times (see details in Materials and Methods). Cracked samples were randomly chosen and exposed for 15 min to a light of λ > 280 nm for crack healing. Figure 4a shows the microscope image of a damaged specimen with a wide and deep crack. After exposure to UV light, the crack size was visibly decreased and some parts had disappeared or closed up (Figure 4b). This was more dramatically confirmed by the fluorescence images observed from the same area (Figure 4c,d). In Figure 4c, the obvious fluorescence is observed from the damaged area; there is even a long diagonal crack from the top left to the center, which was obscure in the optical microscope image. This indicates that the TCE-co-DCE monomers from the microcapsules filled up the gap of the cracked area. However, in Figure 4d, the fluorescence signal is completely disappeared from the substrate, meaning that the TCE-co-DCE monomers were polymerized. The polymerization of TCE-co-DCE monomers apparently healed the damaged area. This result shows that our system of core-shell microcapsules successfully worked as a self-healing agent.

### 3.4. Repeatability of Self-Healing

The biggest disadvantage of encapsulation self-healing is that it only serves for one-time healing, in addition to the discontinuity due to the difference between the core therapeutic agents and the shell of the microcapsules [13,14,15,16]. In our above result, we demonstrated that this discontinuity could be solved by the core-shell microcapsule of poly (TCE-co-DCE) containing TCE-co-DCE monomers. Now, we prove the repeatability of our system. Figure 5a–e schematically show the repeatable healing process of our system. Figure 5a shows the core-shell microcapsule before cracking, which corresponds to the #1 FT-IR spectrum of Figure 5f observed from the microcapsules. The characteristic polymer peak at about 1734 cm^−1^ is clearly seen. When the specimen is cracked by external impact or internal fatigue, embedded microcapsules spatially close to or on the spot of cracking could be broken by stress (Figure 5b). The monomers inside the microcapsules leak out and fill up the crack. In this process, the FT-IR peaks at about 1713 cm^−1^ and 1637 cm^−1^ are observed as seen in the #2 spectrum of Figure 5f. Figure 5c shows the temporary cross-linking of TCE-co-DCE molecules under irradiation, which is reflected in the #3 spectrum of Figure 5f, the peak at 1637 cm^−1^ disappearing again and peak shifting to 1734 cm^−1^. If crack occurs again in this material, the cyclobutane rings will be cleaved and the remaining monomers will leak out. However, in practice, it is hard to detect the change in the chemical structure of a photo-cross-linked sample while propagating the microcracks using FT-IR, so we ground the sample in a mortar for 15 min to make a fine particle. This helps the cyclobutane rings convert into cinnamate moieties, resulting in the detection of structure change by FT-IR, as seen in the #4 spectrum in Figure 5f. In the #4 spectrum, measured from the ground sample, both the cinnamoyl C=O (as a shoulder at 1713 cm^−1^) and C=C (at 1637 cm^−1^) bands appeared, indicating the reversal of cyclobutane cross-links to the original cinnamate. On the basis of the FT-IR analysis, it is estimated that 16% of the cyclobutane rings were transformed into cinnamoyl C=C bonds after grinding. However, this cinnamate absorption disappeared again after UV re-irradiation of the ground sample (Figure 5e and #5 spectrum of Figure 5f). This suggests that only the cyclobutane cross-links were disconnected (to form cinnamoyl C=C bonds) and that the other bonds were almost intact when the sample was ground. The exclusive cleavage of cyclobutane is attributed to its lowest bond strength (due to the high ring strain) compared to that of all the other bonds. Based on this result, we can expect the repeated cycle of the exclusively cleaving cyclobutane while microcracks form and propagate in the photo-cross-linked sample, leading to the recovery of cyclobutane rings through [2 + 2] cycloaddition between cinnamoyl C=C bonds under irradiation of light. Moreover, the photochemical healing process we developed occurs very quickly and does not require any catalyst, additive or severe heat treatment.

## 4. Conclusions

In this study, we synthesized a core-shell-like microcapsule as a self-healing material, where the core and shell were composed of the same base molecule, TCE-co-DCE. Using this core-shell-like microcapsule, we demonstrated the repeatable self-healing process along with the interface continuity between the core healing agent and the carrying shell material. The underlying mechanism is the cross-linking of TCE and DCE via [2 + 2] photocycloaddition, the cleaving of cyclobutane cross-links into the original cinnamoyl groups under stress and impact, and the re-photocycloaddition of cinnamoyl groups into cyclobutane rings. These were confirmed by FT-IR spectra measured at each step. The self-healing process we developed is believed to provide a promising way to overcome the disadvantages of the microcapsule method.

## Figures and Tables

**Figure 1 polymers-11-00104-f001:**
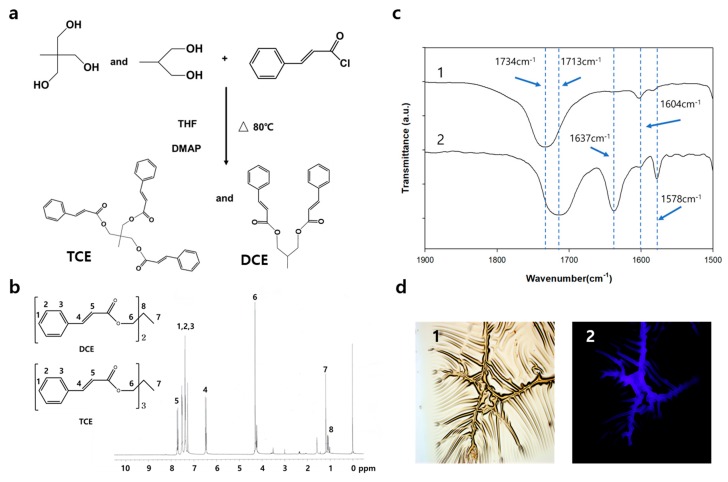
Synthesis and analysis of TCE and DCE monomers. (**a**) The molecular structures of precursors, TCE and DCE after synthesis. (**b**) ^1^H NMR spectrum of the TCE and DCE co-monomer. (CDCl_3_, 300 MHz) 1.17 (3H (CH_3_) and s), 4.29 (6H (3 × OCH_2_) and s), 6.45 (3H (3 × phCH) and d), 7.33 7.51 (15H (aromatic), m), 7.70 (3H (3 × phCHCH) and d). (**c**) Comparison of FT-IR spectra of poly (TCE-co-DCE) film without (1) and with (2) exposed monomers. (**d**) Optical (1) and fluorescence (2) photograph images of poly (TCE-co-DCE) film.

**Figure 2 polymers-11-00104-f002:**
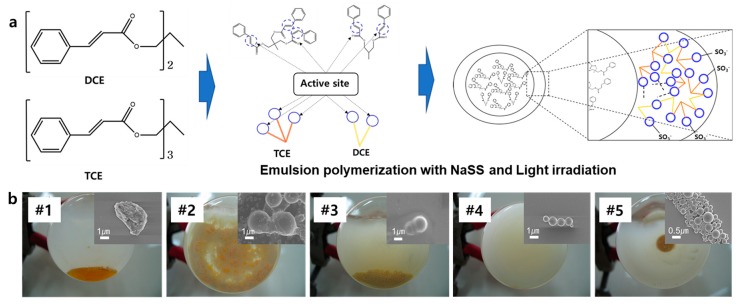
Core-shell-like microcapsules of TCE-co-DCE. (**a**) Schematic process of polymerization and fabrication of microcapsules. (**b**) Digital images of TCE-co-DCE particle emulsion in the flask, synthesized with different ratios of TCE to DCE. #1: 90, #2: 70, #3: 50, #4: 30, #5: 10. Inset: SEM images of microcapsules.

**Figure 3 polymers-11-00104-f003:**
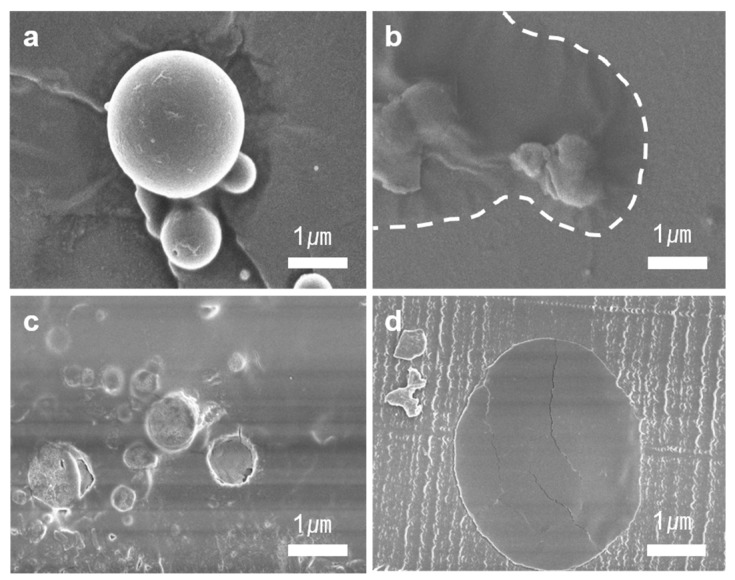
Cracking and crack-healing of s microcapsule. SEM images of (**a**) micro-sized core-shell-like TCE-co-DCE capsules; (**b**) TCE-co-DCE monomers spilled out of the cracked microcapsules on to the floor; (**c**) microcapsules embedded in an epoxy resin; and (**d**) a microtome sectioned area after cracking and irradiation of light.

**Figure 4 polymers-11-00104-f004:**
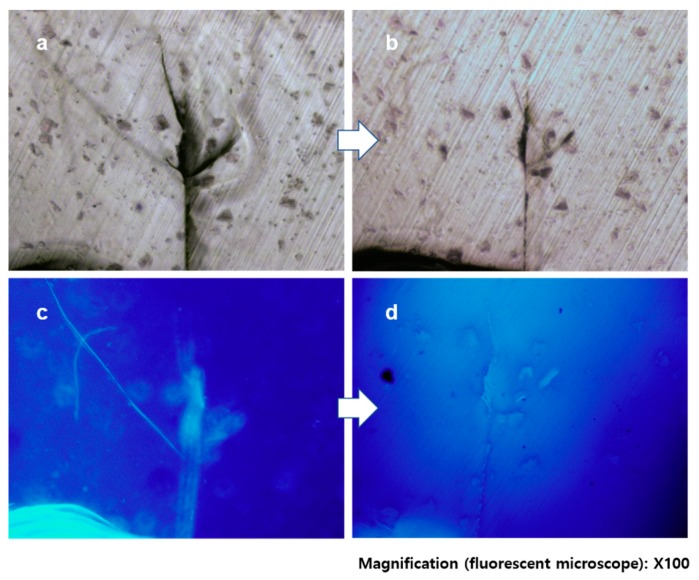
Crack-healing of TCE-co-DCE. (**a**,**b**) Optical images of cracked epoxy embedding self-healing microcapsules (**a**) and the same area after irradiation of light (**b**). (**c**,**d**) Fluorescence images of cracked epoxy (**c**) and the same area after irradiation of light (**d**).

**Figure 5 polymers-11-00104-f005:**
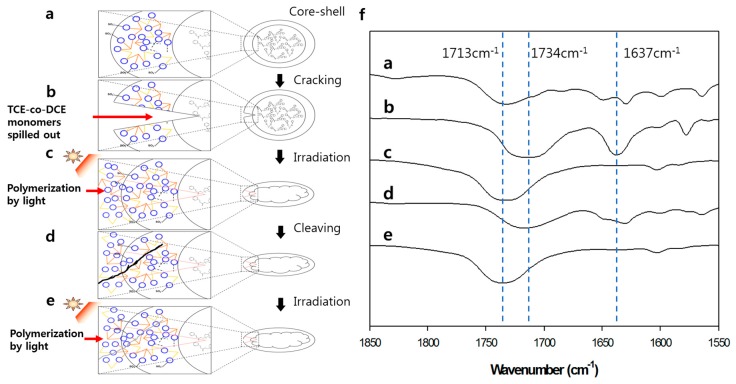
Repeatability of self-healing. (**a**–**e**) Schematics of core-shell-like microcapsules to repeatedly heal the crack. (**f**) FT-IR spectra for each step of repeated cracking and healing processes shown in (**a**–**e**).

**Table 1 polymers-11-00104-t001:** Viscosity value by monomer ratio.

Monomer Ratio (TCE:DCE)		Temperature (°C)	Viscosity (cP)
TCE		22	12500
#1	90:10	21	8380
#2	70:30	21	6030
#3	50:50	21	5703
#4	30:70	22	4457
#5	10:90	21	1025
DCE		21	808

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
