# Peer review of "Repeatable Crack Self-Healing by Photochemical [2 + 2] Cycloaddition of TCE-co-DCE Monomers Enclosed in Homopolymer Microcapsules"

_polymers, 2019, doi:10.3390/polym11010104_

Reviewer 1 Report

The paper is very interesting as deals with an important topic.

It is necessary to give information about the process how the microcapsules were embedded into the epoxy.

Did the Authors make any experiments when the healing components were directly involved into the coating, i.e. not in the form embedded into microcapsules? When there are results, it could complete this interesting work.

Special remarks

Page 1 line 19: ”embedding”

Page 1 line 19: ”ovious” Not “obvious”?

Page 1 line 32: “autonomically” It is not better to use “autonomously”?

Page 1 line 42: “stimuli

Page 2 line 44: ”propagating”

Page 2 line 46:”microcapsule”

Page 2 line 47:”originalstate;”

Page 2 line 51:”cannot”

Page 2 line 57: (TCE) and (DCE)

Page 2 line 59: „the homopolymer shell made of TCE and DCE.” Is it a homopolymer? Not a co-polymer?

Page 2 line 62:”specimens, which…”

Page 2 line 90: “distilled and deionized”

Page 3 line 92: What was the wave length of the UV light?

Page 5 line 173:  specimen

Page 5 line 175: “shows the micro toming cross section” Please, rephrase this part of the sentence.

 “Figure 4. Crack-healing of TCE-co-DCE. (a and b) Optical images of cracked epoxy embedding self- healing microcapsules (b) and the same area after irradiation of light (b).” It would be better to mention: Crack-healing by TCE-co-DCE, and make it clear, which is the (A) and (b).

Author Response

 We thank the reviewers for their valuable comments and kind suggestions. We are also glad to send this clarification report to the editor and reviewers. Our manuscript has been explicitly revised in accordance with the reviewers’ comments and suggestions, and the modifications are marked with red color in the revised manuscript.

Reviewer 2 Report

The present manuscript entitled ”Repeatable Crack self-healing by photochemical [2+2]  cycloaddition of TCE-co-DCE monomers enclosed homopolymer microcapsules” by S. Kim et al. describes the fabrication of microcapsules based on cinnamoyloxymethyl-based monomers. The monomers were utilized for the fabrication of microcapsules via photocrosslinking of the liquid monomers. The monomers in the core are not polymerized. These microcapulses have been utilized for the fabrication of epoxy composites. The crack healing was investigated. Overall the study is interesting; however, some open issues / questions remain:

The capsules are described as “homopolymer microcapsules”. However, two monomers are used TCE and DCE. Copolymer would be a better description.

NaSS should be named NaPSS

How were the microcapsules purified / isolated? The description of the synthetic procedure is rather short.

The synthetic procedure for the TCE and DCE is confusing. The mixture was utilized for the synthesis. However, the monomers were isolated afterwards. Why was the mixture utilized and not the single compounds.

“Using the TCE-co-DCE monomers, the poly (TCE-co-DCE) was synthesized and then confirmed their chemical structure using 1H-NMR spectrum as shown in Fig. 1 (b).” The sentence is confusing. The depicted NMR spectrum shows the monomers and not the polymer.

How thick is the shell of the particles? How much liquid monomer is left inside the capsule.

The released monomers could be polymerized by UV light. How deep is the UV light penetrating the specimen? In the synthesis of the particles the penetration depth is limited, which allows the synthesis of core-shell particles. During the healing event this penetration limit will be limiting the healing performance.

The mechanochemical activation is still questionable. An incomplete polymerization (due to the limited penetration depth) would also lead to similar observations. The monomers are polymerized at the surface and will leak later on when a second scratch occurs or when the materials is grinded.

Author Response

(The authors gave the same response as above.)
